# Facebook as an engagement tool: How are public benefit organizations building relationships with their public?

**Marian Olinski** *, **Piotr Szamrowski**

Institute of Management and Quality Sciences, University of Warmia and Mazury in Olsztyn, Olsztyn, Poland

☯ These authors contributed equally to this work.
* olinski@uwm.edu.pl

**Data Availability Statement:** All relevant data are within the manuscript.

**Funding:** The authors received no specific funding for this work.

## Abstract

The beginning of the 21st century brought a change in the approach to public relations (PR). Both the literature and the practice saw a shift in focus from evaluating the efficiency of communication strategies to relationship management. On the other hand, the growing interest in the use of social media in the management of many types of organizations has prompted researchers to seek the theoretical causes of this trend. Therefore, the purpose of this research is to examine the scope of the Polish public benefit organizations' (PBO) use of the social network Facebook in managing relationships with stakeholders. The PBO is a specific form of non-profit organization, which enjoys the special privilege of collecting tax-deductible donations of 1% of personal income tax (PIT). The research covered 876 entities, which were divided into four clusters depending on their size. A database of posts was created that the organizations published in the period selected for the research. Then, the impact of the Facebook content on user engagement was analyzed. For this purpose, various statistical methods were used, i.e. descriptive statistics, statistical tests and multiple regression analysis. The results indicate that, in spite of social media's unquestionable advantages, public benefit organizations only take advantage of a small proportion of this potential. The role of the most popular service, Facebook, in increasing organizational revenue from 1% of PIT deductions seems secondary at best. Apart from that, the results showed that the rates of posting by different Polish public benefit organizations vary widely. Although some organizations were very active in this aspect, the Facebook pages of many organizations remained completely inactive throughout the analysis or showed only minimal activity. Larger organizations exhibited a much greater posting activity than their smaller counterparts.

## Introduction

The role of PR is no longer limited to the dissemination of information but has been extended to building positive relationships with the organization's stakeholders. Managing those relationships has become the key paradigm in PR theory, and the relationships between the organization and its stakeholders has become a central part of its practice. Social media services are

**Competing interests:** The authors have declared that no competing interests exist.

an especially useful instrument in this process since they offer a vast array of means facilitating interaction with the stakeholders of an organization. Their characteristics, such as egalitarianism, interactivity, low cost of access, and ease of use, form an excellent environment for building relationships based on dialogue. These characteristics are especially desirable for organizations engaged in non-profit activity, since they commonly do not have large budgets, which makes reaching a broad group of recipients with specific values and ideas a relatively challenging task. Hence, actively using social media may be an effective tool in the realization of that goal, enabling access to traditional media, potential donors, volunteers, etc.

The inspiration for investigating issues connected with social media use in public relations management originally came from the subject of the research itself; for example, a public benefit organization and its unique method of financing its activity. A public benefit organization is a specific form of a non-profit organization which enjoys the special privilege of tax-deductible donations of 1% of PIT (personal income tax). This fact makes it especially worthwhile for these organizations to use as many channels of communication as possible in their activities, including online channels.

Research conducted on the organizational uses of social media has primarily been concerned with commercial, for-profit entities. The scholarship literature on non-profit organizations is much sparser. This is a surprising fact, since social media can be very useful in several fields, e.g. volunteer recruitment [1, 2], fundraising [3, 4], media exposure [5] and especially relationship management with the stakeholders of the organization [6]. The literature analyses user engagement with contents published primarily on Facebook and Twitter [e.g. 7, 8], but the number of studies is low since the term engagement itself is not entirely understood, which impedes the analysis. Previous research has also concentrated on large organizations, primarily based in the United States and, again, their organizational uses of Facebook and Twitter, which is especially popular in Anglo-American countries ([8–15]). Moreover, the researchers of these issues, when choosing a research sample, most often limited themselves to the method that can be described as "top-100-style sampling". It creates a research gap, as noted by Nah and Saxton [16], denoting the underrepresentation of small- and medium-sized organizations in research samples. The use of random sampling is the solution to this problem, allowing more accurate inference via increased generalizability and simpler hypothesis testing. The appeal of both of these researchers only in a few cases was taken into account [16]. Polish academic literature regarding the utilization of social media in the activities of non-profit organizations is also very modest. With a few exceptions, there has been practically no research conducted on the role of social media in building relationships with the community by non-profit organizations, especially by public benefit organizations [e.g. 17–19].

The analysis of academic literature also indicates that the vast majority of studies conducted so far have dealt with only a single social media channel [20]. Therefore, there is a lack of research describing the coordination process of different communication channels. It seems that this issue is particularly important from a practical point of view, enhancing the stakeholder's engagement.

The present article aims to fill in at least part of that gap. Firstly, it contributes to the literature by examining the scope of the public benefit organizations' use of the Facebook social network in managing relationships with the public. This study analyzed the following question: how do the selected organizational parameters, available communication tools used in a single post, and basic Facebook profile features influence the public engagement rate? The findings may offer suggestions for non-profit organizations operating in other countries, since the role of social media in building relationships with partners and the public is becoming increasingly important around the world (not only in Europe and the US). This is all the more important for non-profit organizations, since most of them have limited resources, which they prefer to

utilize on the realization of their mission, rather than on the costly process of building and maintaining relationships with users. Secondly, the research described in the following article was part of a larger research project which included an analysis of the use of Twitter in the activities of Polish public benefit organizations [17]. The research methodology used for its analysis was then applied to Facebook research. The general conclusions formulated on this basis made it possible to identify differences in the organizational use of both communication channels and the development of practical guidelines for coordinating the utilization of different social media services in a non-profit organization. Thirdly, random sampling was used to select the organization for the research, taking into account not only large organizations, but also medium and small organizations, meeting the appeal formulated by Nah and Saxton [16].

## Literature review

### Relationship management paradigm in the theory of non-profit organizations

The growing interest in the use of social media in the management of many types of organizations, from strictly commercial to political and social enterprises, has prompted researchers to seek the theoretical causes of this trend. The most common avenue has been through existing theories, attempting to explain the organizational uses of social media in established terms. One of these seem particularly useful in delineating the theoretical underpinnings of the dynamic development of media presence among both commercial and non-commercial organizations. It is known as resource dependency theory and is based on the assumptions of stakeholder theory as well as the relationship management paradigm of public relations, which is currently the dominant approach in PR [21, 22]. Resource dependency theory is a starting point in explaining the reasons for the use of social media in the activities of a non-profit organization. As indicated by Mato-Santiso, Rey-García and Sanzo-Perez "this is because of non-profit organization's endemic lack or shortage of resources relative to the size of their target groups' needs (potential recipients, beneficiaries or users) and to the complexity of the social problems tackled" [20]. The employment of these theories in attempts to explain the organizational employment of modern communications technologies can be found in the works of such authors as Xu, Saxton, Guo, Baird, Furukawa, Raghu, Guidry, Messner [7, 23–25]. Stakeholder theory assumes that the activity of an organization can be understood as a network of connections with entities that are interested in the results of its activity [26]. The organization is therefore treated as inextricably linked with its stakeholders. Stakeholders are partners in this system in pursuit of organizational goals. Stakeholder theory offers many insights to managers, especially in establishing for whom the organization is creating value, indicating the means with which that value is to be created, monitoring the behavior of the organization's stakeholders to improve the processes within the organization and maintaining ethical standards while creating value for the organization's stakeholders. The implementation of these four elements should create a hierarchy of importance among the organization's interest groups and indicate potentially effective tools of influencing them. Public relations techniques may be helpful in doing that effectively. The dominant approach in this area is relationship management, which is a development of the resource-based view that inter-organizational relationships are a strategic, idiosyncratic and invisible resource of an undertaking. However, this is connected with the evolutionary approach and business ecology, which maintains that organizations are not isolated entities, but they are anchored in systems and networks of relationships which can be shaped through emergent means and self-organizing mechanisms. Adopting the relationship paradigm means that public relations are to be treated as the management function that establishes and maintains mutually beneficial relationships between an

organization and the public, on whom its success or failure depends [27]. Assessment of these relationships is of key importance since it is beyond doubt that they should be positive if they are to indirectly enable the realization of the entire organization's goals. The normative approach dictates that in building positive relations with stakeholders, the organization should use such communication strategies which trigger public engagement.

## Digital engagement in public relations literature

The research that has investigated the use of social media for PR activities of non-profit organizations has so far been focused on only a few of its aspects. First, on the scope of social media usage and on which characteristics employed by the NGO increased the use of those services [e.g. 16]. Second, static elements of an organization's profile on the social media site were analyzed [24, 28]. Third, the perception of the role of social media for an NGO by public relations practitioners were examined [e.g. 29]. Fourth, the contents published on social media sites and their impact on user engagement levels were analyzed [e.g. 12, 14, 30–34]. From the perspective of this article, the fourth aspect mentioned above is the most important which is associated with building public engagement.

Engagement is defined as a dynamic multidimensional relational concept including psychological and behavioral features of involvement, connection, interaction, and participation designed in a way to enable the achievement of goals at an individual, organization or, social level [35]. Engagement can be understood as a process in which engagement is created, or cocreated, through communication or as a state or outcome that encompasses cognitive, affective, and behavioral dimensions [36]. In the online environment, the theoretical foundations for the formation of engagement can be found in the work of, for example, Dhanesh [37], who also understands this concept as a cognitive, emotional and behavioral state in which the organization and the entities in its environment, sharing common interests in areas relevant to themselves, enter interactions which can be placed on a scale. On one end are passive recipients, whose level of engagement is restricted to the cognitive stage, and, to a greater or lesser extent, prompts emotions in them as a result of the absorbed contents. On the other end of the scale are recipients whose engagement level is not restricted to these two stages, but enters the phase of behavior generation which is detectable by the organization. For example, this can be as little as "liking" contents posted by the organization. According to Dhanesh, this type of engagement is most crucial to the organization since it can foster its most vital objectives. In Dhanesh's model, most social media users are passive recipients, for whom the absorbed contents do not evoke an emotional reaction strong enough to prompt a behavior which is detectable by the organization. By this assumption, it seems key for an organization to create the widest possible base of entities with an interest in the organization's mission. In the case of Facebook, that would be, for example, the number of users "following" the profile. The larger the number of such entities, the greater the chance of attracting so-called Social Media Influencers (SMIs), who are individuals most engaged with the organization's activities. Their role is not limited to the passive absorption of content broadcast by the organization, but also extensive commentary, which is shared on the network.

The literature indicates that public engagement is influenced by a number of factors that may result from both the nature of the published content and the very characteristics of a nonprofit organization. The first group of factors may include, for example, the function performed by the published content research conducted by Lovejoy and Saxton [38] which made it possible to create a typology enabling the identification of the basic and specific function of content published in the online environment. This particularly concerned the Twitter content of American largest non-profit organizations. Other studies indicate the importance of

communication tools used in single posts or tweets [e.g. 13]. It may be assumed that the more complex the post, the more likely the recipient would react (the presence of a video file, photos, links to external sites, hashtags or mentions in published content). Moreover, the length of the post or tweet and the time of its publication, e.g. a weekday or weekend, may also translate into a greater level of public engagement. The second group of factors concerning the features of the non-profit organization itself may also influence the public engagement of the content published on social media sites [e.g. 17, 39]. These features may include the value of the organization's assets or annual revenues, its age, type of activity, the number of followers of its social media profile.

In public relations research, the level of engagement is most commonly measured as the number of likes on a post and its shares and comments. As explained by Saxton and Waters [33], such a criterion is convenient since this data is publicly available, does not require consent from the profile's administrator to be accessed and is highly useful and illuminating. Kim and Yang [40] treat pressing a "like" button as an affect-driven behavior, commenting on content as a cognitively triggered behavior and sharing as a combination of both. For example, the recipient of the content, by clicking on the "like" button, informs that he or she has a positive attitude towards the content and identifies with it to some extent. According to Saxton and Waters [33], liking a single post is the weakest form of engagement, as opposed to the number of comments, which should be prioritized from an organizational point of view, reflecting a significant commitment of the content's recipient.

## The essence and characteristics of a public benefit organization

Public benefit organizations are non-governmental organizations (associations and foundations, as well as joint-stock companies and limited liability companies not operating for profit). Public benefit organizations (PBO) constitute approximately 10% of all registered third-sector organizations in Poland (as of November 30, 2020, the list contained 8,732 active PBOs).

The condition for obtaining the status of PBO is that the organization:

- carries out activities for the benefit of the entire community or a specific group distinguished due to a particularly difficult life or material situation in relation to the society (specific purpose of the activity);

- does not conduct business activity or conducts it in sizes serving the implementation of statutory goals;

- all income is allocated to public benefit activities;

- has a statutory collective control or supervision body, separate from the management body, which is not subject to it in the scope of internal control or supervision;

- has a statute, which provides for restrictions on the use of the assets of the organization and purchases from entities associated with the members of the organization;

- subject to entry in the appropriate register—in accordance with the Act on the National Court Register;

- draws up an annual substantive report on its activities and makes it public (on the website of the office supporting the minister responsible for social security);

- submits the above-mentioned reports to the minister responsible for social security. If he does not submit the report—it will not be included in the list of public benefit organizations entitled to 1% income tax. It is this privilege, i.e. the possibility of obtaining 1% of personal

income tax, that is the most important in the functioning of PBOs (in 2019, 8.8 thousand PBOs received a 1% PIT deduction).

Taxpayers have the option of donating 1% of their annual PIT to a specific public benefit organization, which creates a peculiar market. Organizations entitled to receive 1% of PIT donations compete in this market, money being the medium of exchange. The taxpayer transfers it to the Treasury, but as a result of their individual and autonomous decision, it becomes addressed to a specific recipient, and the appropriate tax office is required to pass it on to the indicated organization. The fact that such an organizational format functions in only a handful of countries, mainly in Central-Eastern Europe (e.g. Hungary, Slovakia, Romania and Lithuania), makes even a rudimentary description of its functioning a valuable and new scholarly interest.

## Hypothesis development

Scholarly literature has indicated the great variety in the organizational uses of social media sites. On the one hand, there is a large group of non-profit organizations which use such sites practically every day–on the other, there is an equally numerous group of organizations which exhibit a minimal activity on those sites. It is assumed that because of difficulties with acquiring and retaining qualified employees, the Facebook activity of Polish PBOs is low. This assumption also results from the research conducted earlier by the authors on the use of Twitter by Polish non-profit organizations [17]. Secondly, organizational size measured as total annual revenue has a statistically significant impact on their activity. The importance of this factor is also indicated by the results of the Wallace and Rutherford studies, although the research sample in this case was only 160 non-profit organizations [39]. Organizations with larger budgets usually have better access to qualified staff, which should translate to more effective use of social media sites in their operations. Apart from that, making use of social media sites in an organization's activities is only seemingly cheap since it requires staff responsible for managing the sites to spend a significant amount of time and effort. This issue is often overlooked since devoted time and effort are difficult to appraise. Small non-profit organizations, conducting public benefit activities using a small team of a few people, may encounter serious difficulties with finding a sufficient amount of time to properly utilize social media in the fulfilment of their goals. Hypothesis H1 therefore states:

*H1*: *The scope of Facebook publishing activities by Polish public benefit organizations is influenced by the amount of a given organization's annual revenue, i.e. higher revenue correlates with higher publishing activity on the site.*

Facebook offers a wide set of tools for building dialogue and interaction with a potential audience. The most popular of these are hashtags and mentions. The former serve to highlight and categorize information, the latter serve to direct the message to a particular user or users. Apart from that, profile administrators may add hyperlinks to the text of the contents, append one multimedia file per post (e.g. a photo or video file), or share content published by other organizations and individuals, while maintaining the original form or modifying the posts. It is hypothesized that a higher level of complexity of a post translates to a higher engagement by the recipient. High complexity means that the recipient is going to experience it in multiple ways, which will potentially prompt a particular reaction, e.g. in the form of a comment under the posted material. For instance, posts which contain multimedia files and hyperlinks to external webpages apart from plain text require more engagement to be absorbed by the user. This, in turn, may prompt a reaction desirable from the point of view of the organization. On

the other hand, posts which contain mentions or hashtags may also prompt a recipient's reaction by engaging the reciprocity rule. Hypothesis H2 therefore states:

*H2*: *The level of complexity of the post affects public engagement, i.e. posts enriched with more communication tools have greater public engagement.*

The first research into the function of published content was conducted by Lovejoy and Saxton 38 in a study of the 100 largest American non-profit organizations, as defined by total revenue. They identified three basic functions that published content serve: information, community building and call to action. The goal of previous studies was not only to identify the functions of published content, but also to identify their impact on user engagement. For example, Saxton and Waters [33] indicated that community-building content around the organization's goals generates more likes and comments than informational content, which, in turn, are shared more than content with another basic function. A 2016 study of healthcare-related organizations, mainly involved in HIV prevention and treatment of AIDS [31, 41], also found that the function of published materials impacts the user's level of engagement. Hypothesis H3 therefore states:

*H3*: *The basic function of content significantly impacts public engagement. Posts with the leading community building function have the greatest impact.*

Hypotheses H4 and H5 analyze the impact of selected characteristics of an organization and its Facebook profile on social media public engagement. Two organizational characteristics were taken into account: size, measured by total annual revenue and annual intake of 1% of PIT contributions. In the case of Facebook profiles, the parameter was the number of observers. The following causal relationship was hypothesized: in accordance with resource dependency theory [21], non-profit organizations whose 1% of PIT contribution dependency is significant will concentrate more heavily on active employment of social media sites as a means to deliver their ideals and values to the broadest possible audience, who, in this case, are very dispersed. The literature also indicates that organizational size, measured by annual revenue, carries with it better abilities to effectively employ available computer technologies [e.g. 42]. An organization's use of social media sites is only seemingly cost-free. It requires, first of all, hiring competent staff (which is usually costly), as well as devoting much of their time to the ongoing management of those channels. For example, Baird *et al.* [24] point out that the limited financial capabilities of smaller organizations cause a gap in the use of modern computer technologies between the smaller and larger organizations to widen continuously. What is more, as an organization grows, it becomes more visible not only offline, but also online, attracting attention from individuals, local and state authorities, or from the media [16]. An organization's size may thus translate to a higher overall user engagement rate with contents published on social media.

The present author's studies on the role of Twitter in PBO activities [17], indicate that the number of users observing an organization's profile on this site has an even greater impact on user engagement than the size of the organization as measured by total annual revenue or the value of its 1% of PIT revenue. From an organizational point of view, the key concern is thus to cause as many users as possible to "follow" its profile. The higher that number, the greater the probability that the posted contents will evoke a reaction, generating a so-called snowball effect, and delivering it to a broad group of recipients, who may, in turn, join the profile's followers. It seems that organizations which are large, in terms of their total revenue, have greater abilities to create a sufficiently large follower base, since they can utilize both offline and online channels for that purpose. In addition, it must be emphasized that this should be the priority for organizations relying heavily on contributions from individual donors (e.g. 1% of PIT

contributions). Smaller organizations, due to their financial limitations, may have difficulties creating a follower base of sufficient numbers, rendering their output unlikely to reach a wide audience and generate desirable reactions.

Hypotheses H4 and H5 therefore state:

H4: *The higher the number of followers of an organization's Facebook profile, the higher the user engagement.*

H5: *The number of followers is more significant for the user engagement level than the organization's size as measured by total annual revenue, amount of 1% of PIT contributions and employment.*

## Research methodology

### Sample

Sample selection proceeded in the following steps:

1. Identification of the organizations (checking whether the organization still has the PBO status);

2. Analysis of the technical and financial reports of public benefit organizations from the database available on the website of the National Freedom Institute. As of 1 January 2017, it contained 8,250 organizations, 408 of which were eliminated from the study, as they either entered the process of dissolution, or failed to file their financial and technical reports. The sample thus contained 7,842 organizations. The following information was collected from the database: the organization's geographic service area, the main type of public benefit activity, total revenue, revenue from unpaid and paid public benefit activity, commercial revenue, 1% of PIT revenue, revenue sources, number of full-time employees (or equivalent) and civil-law contracts, total amount of remunerations divided into full-time positions and civil-law contracts, number of members divided into natural and legal persons (in the case of associations) and the extent of reliance on volunteer work (either by employees or the public).

3. Identification of the most utilized social media site to be further investigated. As a first step, it was verified that the non-profit organizations in the study had their own websites. In most cases, organizations would include the address of such a site in their technical report. In other cases, the website was identified with the internet search engine google.pl. This allowed us to identify the 67% of the organizations which had their own website. The scope of each organization's online engagement was analyzed mainly through those websites, but also with the google.pl search engine, since some of the organizations only used social media sites for their operations, without making use of their own website.

The social media site which the studied PBOs primarily used was Facebook, and further research focused on it (3,100 Polish PBOs, i.e. almost 40%, used it in their operations, with YouTube a distant second at 5.6%). The public benefit organizations (N = 3100) which administrated their own fanpage (a profile, or Facebook page which is an official showcase of an organization) were categorized in four groups depending on total annual revenue in 2017, namely: group I–revenue of PLN 10 million and above (77 organizations), group II–revenue from PLN 1 million but less than 10 million (555 organizations), group III–revenue from PLN 100 thousand but less than 1 million (1384 organizations), group IV–revenue less than PLN 100 thousand (1,084 organizations).

The minimum sample size for a finite population (for a confidence level of 0.95 and maximum error of 0.05) with the groups (strata) thus defined was calculated to be 876

subjects (64 from group I, 227 from group II, 301 from group III, and 284 from group IV). The PBOs from each stratum were sampled with the Research Randomizer algorithm.

## Data collection

The next stage of the research included the construction of a post database. Between 15 May and 12 July 2019 (90 days), two specific days of the week were randomly selected–two Mondays, two Tuesdays, etc. (barring any two selected days from being adjacent). The utilization of such a day selection procedure was aimed at eliminating events which would result in atypical, sudden peaks in the activity of an organization (hence the choice from a 90-day period). The Facebook pages of all 876 organizations on each of the 14 selected days were checked and each post was entered into the database to be analyzed (the entire database amounted to 4,621 posts). The analysis first involved identifying the length of a post (number of characters), date of publication (weekday or weekend) and type of post (completely original content, modified shared content, unmodified shared content, live feeds, a post adding pictures to an album or a post announcing an event or a post sharing another page's event). The post's communication tools were then identified, i.e. use of links to external pages, mentions, hashtags–with numbers of each post. The posts with attached images, videos and graphic emoticons were also marked. Then the basic function of each post was identified by analyzing its contents. Finally, the user engagement was recorded with the numbers of likes, shares and comments.

## Data operationalization and analytical procedures

The data from a Microsoft Excel spreadsheet were exported to the Statistical Package for the Social Sciences (SPSS), version 25, where its accuracy and completeness was additionally verified. SPSS served to conduct statistical analysis of the gathered research material. First, the publication frequency rate was analyzed, as well as the utilization of available Facebook communication tools. Both publication frequency and the frequency of the communication tool use were measured separately for the entire sample of 876 organizations, as well as for each total annual revenue group. The data analysis was conducted using univariate descriptive statistics associated with frequency distributions, including percentages, means and standard deviations as well as statistical tests that included Kruskal-Wallis testing, which verified hypothesis H1.

A multiple regression analysis was then conducted to identify variables which best predict user engagement with the posts published by the organization. Such an analysis is especially advised for highly correlated predictors since it allows the introduction of the best-fitting variables for the model or to remove variables that lose their predictive power in a series of successive steps. Because of the significant positive skew of both dependent variables and quantitative predictors, the regression of the variables was performed. Since a logarithmic (log10) regression which can be employed for positive values is recommended for right-skewed analyses, each value was increased by 1 before performing the aforementioned regression. Polytomous qualitative variables were then re-coded using dummy coding, resulting in a number of dichotomous variables denoting the presence or absence of a given parameter. The significance threshold was set at .05.

The following variables were considered as potential predictors to verify hypotheses H2, H3, H4 and H5:

- basic parameters of the organization's Facebook profile, i.e. the number of followers, profile age (in days), and total number of posts in the analyzed 14-day period,

- day of posting (days were distinguished into weekend days, i.e. Saturday and Sunday, and weekdays–from Monday to Friday), post length, and its type,

- communication tools used in the Facebook contents, i.e. number of hyperlinks to webpages of other organizations or other pages of the organization in question, number of mentions and of hashtags in the post, and the presence of images or videos,

- the basic function of the given post,

- selected characteristics of the organization, i.e. amount of annual revenue, revenue from 1% of PIT, revenues from gratuitous public benefit activities (free-of-charge services for beneficiaries), revenues from business activities, paid public benefit activities for beneficiaries (although the beneficiary payment service covers only the costs without profit for the organization), revenues from private sources, number of employees with full-time position equivalent, type of main public benefit activity, and the geographic service area.

For employment, the studied organizations were divided into four groups, i.e. organizations which, in full-time position equivalent, employed more than 50 people in the course of the year (N = 64), those which employed from 10 to 50 people (N = 126), those which employed at a level from 0.1 to 10 positions (N = 243), and those organizations which had no employees (N = 443).

Based on the type of main public benefit activity, the PBOs were divided into 5 groups, i.e. organizations involved with the economy and improving economic development (N = 19), those involved with social policy and social aid (N = 316), healthcare (N = 96), arts and culture, sports, education, and development of civil society (N = 425), as well as rescue services and public safety (N = 20).

As far as geographic service area is concerned, the organizations were divided into those operating abroad (N = 38), in the entire country (N = 244), in a voivodeship or several voivodeships (N = 142), in a district (*powiat*) or several districts (N = 98), in a commune (*gmina*), several communes, or in a neighborhood (N = 102). In this case, the study did not encompass the whole sample, but only 624 organizations (the remaining 252 organizations did not include a geographic service area in their technical reports). For coding the basic function of posts, the scheme proposed by Lovejoy and Saxton was employed [38]. Their studies distinguished three basic functions performed by posts: informing, community building, and call to action. Information posts are those primarily concerned with educating the user, or informing about the activities of the organization. They are also posts which serve "dry" facts, such as informing that the organization's office will be closed or will close early on a given day. An organization which publishes primarily informing posts is devoting itself mostly to one-way communication. Community-building posts aim to create a positive relationship with the organization's stakeholders and to engage them with the organization's cause. Posts with this function are key to building dialogue-based communication. Finally, posts which are calls to action are intended to prompt the user to do something for the organization's benefit. Compared to informing or community-building posts, these have the most tangible effect. For example, a page administrator can relatively easily judge the effectiveness of posts which solicit donations to the organization, especially if they contain a link to a payment service.

The dependent variables for hypothesis H2, H3, H4 and H5 were the number of likes, comments, and shares which measure public engagement (with total engagement being the sum of all three reactions). Liking suggests the post is appreciated by the public; commenting is a way of responding to a post and building a relationship with the organization; and finally, sharing enables users to actually disseminate the messages to their networks, which would then boost the exposure of the shared content among a broader public. Total engagement was measured

as the sum of likes, shares, and comments. First, the whole database of posts (N = 4621) was analyzed, and subsequently divided them into four revenue groups: under PLN 100 thousand, from PLN 100 thousand to under 1 million, from 1 million to under 10 million, and at or above 10 million.

## Results

The results of the study indicate that the publishing activity of Polish public benefit organizations in the studied 14-day period was low and highly variable, as indicated by the value of the standard deviation (SD = 7.38). Collectively, the studied organizations (N = 876) published 4621 posts, an average of 5.28 posts per subject. Only 19 organizations were identified which published at least 2 posts per day (a minimum of 28 posts in the 14 days). The record was held by an organization which published 51 posts over that period, which means it published an average of 3.64 posts per day. For a large portion of the subjects, the Facebook pages remained completely inactive during that time, or their activity was minimal. As many as 254 organizations were identified (29.0%) which, in spite of having an organizational Facebook page, posted no content over the 14 selected days. Due to the mean's sensitivity to extreme values, a more appropriate indicator seems to be the median, which reached 2.5 published posts per selected day. ¼ of the organizations published nothing on those days (25th percentile), and ¼ published at least 7 posts (75th percentile).

The Facebook posting rate was less for the organizations with lower total annual revenue. By this measure, the organizations were grouped. Group I—published an average of almost 9 posts in the selected 14 days (568 posts in total), group II– 7 posts (1610 posts in total), group III– 5.58 posts (1681 posts in total), and group IV– 2.68 (762 posts in total). The values of standard deviation indicate that publishing frequency was strongly varied within the groups (being 6.93 in group I, 7.62 in group II, 7.92 in group III, and 5.71 in group IV). Among the largest organizations, 5 subjects (7.8%) were identified who published no content on their Facebook pages within the selected 14 days. With decreasing revenues, the numbers of inactive organizations noticeably grew, being 19.4% in group II, and 24.6% and 46.1% in groups III and IV respectively. Notably, the group with revenue above PLN 10 million had only one subject who published at least 28 posts within the 14 days (2 posts a day, with the record holder at 30 posts in total). In group II, six such organizations were identified (record holder at 32 posts), as many as nine in group III (record holder at 46 posts), and three in group IV (record holder at 51 posts). The following median values for publishing frequency were obtained (within the 14 selected days):

- 7 posts in group I; this value reflects that 50% of the organizations published 7 or fewer posts within the 14 days, while 50% published more. A quarter of the organizations published 4 or fewer posts on those days (25th percentile), while another quarter published more than 13 posts (75th percentile);

- 5 posts in group II; ¼ of the organizations published 1 or fewer posts (25th percentile), and ¼ published at least 10 posts (75th percentile);

- 3 posts in group III; ¼ of the organizations published 1 or fewer posts (25th percentile), and ¼ at least 7 posts (75th percentile);

- 1 post in group IV; over 46% of the organizations published no posts within the selected days, while ¼ published more than 3 posts (75th percentile).

The Kruskal-Wallis test additionally confirmed that organizational size, as measured by total annual revenue, had an influence on publishing frequency in the 14 selected days ($\chi2(3)$

**Table 1. Rate of communication tool utilization in Facebook posts by organizational size.**

| Organization size (revenues—PLN) | Number of posts during 14 days (items) | Images (%) | Videos (%) | Links (%) | Emoji (%) | Mentions (%) | Hashtags (%) |
|---|---|---|---|---|---|---|---|
| Over 10 m (group I) | 568 | 82.6 | 12.3 | 57.9 | 53.0 | 29.6 | 20.1 |
| From 1 to 10 m (group II) | 1610 | 87.9 | 8.8 | 47.5 | 48.6 | 24.0 | 12.4 |
| From 100 k to 1 m (group III) | 1681 | 81.2 | 8.9 | 35.2 | 38.3 | 12.7 | 7.9 |
| Below 100 k (group IV) | 762 | 80.3 | 10.9 | 50.0 | 24.0 | 5.8 | 3.7 |
| **Together** | **4621** | **83.6** | **9.6** | **44.7** | **40.9** | **17.2** | **10.1** |

Source: Own data

= 118.772, p<0.001; (average rank per group I = 611.36; average rank per group II = 516.73; average rank per group III = 454.85; average rank per group IV = 319.69). Pairwise comparison showed that the largest organizations by total annual revenue demonstrated higher publishing frequency than those in all of the other groups (p<0.001 for groups III and IV, and p = 0.044 for group II, with a Bonferroni-corrected significance level). Pairwise comparison also indicated that the smallest subjects were characterized by a significantly lower publishing frequency. It was lower not only in comparison to subjects from group I, but also II and III (p<0.001 in each case). Differences in publishing frequency among organizations in group II (annual revenue from PLN 1 million to under 10 million) and III (annual revenue from PLN 100 thousand to 1 million), were also observed in favor of the former.

The H1 hypothesis was thus confirmed. The frequency of post dissemination by organizations with higher total annual revenues is higher compared to those with lower revenues.

The verification of the research hypotheses H2, H3, H4 and H5 was preceded by an analysis of the utilization scale of communication tools available on Facebook (images, videos, links to external sources, emoji, mentions, and hashtags). The lack of such communication tools in the posts would make it impossible to confirm or reject the H2 hypothesis. Frequency analysis indicates that the most used communication tool in PBO postings were images. 3,861 posts from the data sample contained one or more images (83.6%, Table 1). The size of the organization did not influence the frequency of individual posts with images. Only organizations from group II posted such content with a slightly higher frequency (87.9%). About a tenth of the posts contained video (9.6%). In this case, the size of the organization also had no great bearing on the format's frequency of use. Almost 45% of posts contained links to external pages. In the case of this tool, the size of the subject did influence the frequency of use–the largest subjects used it significantly more than other organizations. The difference in utilization is rather large when comparing groups I and II, and very large when comparing groups I and III. The difference in percentage of use as compared between the largest and smallest subjects is noticeably less. The overall average number of links per post was 1.11 and held at approximately that level in all groups. Ideograms denoting emotions, called emoticons or emoji, were used in 40.9% of posts (many applications transform emoticons into images, referred to as emoji, which are commonly referred to as graphic emoticons, while original emoticons are known as character emoticons). Here too, the size of the organization had an influence on the frequency of use. Subjects from groups I and II used them notably more frequently than those from groups III and IV. Mentions and hashtags were decidedly the least used tools, with the former utilized in slightly over 17% of posts, the latter–merely in every tenth (10.1%). Organizational size had a large bearing on the employment of both these tools. The smallest subjects used them to a minimum extent, while the largest subjects took advantage of them much more. Study results show a very definite trend in this regard–the smaller the organization, the less it uses the options of mention and hashtag. The number of uses of these tools per post containing them

was relatively high. On average, a single post contained 2.34 mentions of other entities (introduced with the character @), and 2.96 hashtags (introduced with the character #).

To verify research hypotheses H2, H3, H4 and H5, a multiple regression analysis was performed, initially for all posts, and then segregated by organizational size as measured by total annual revenue. Independent variables (predictors) were composed of a number of factors, connected with the characteristics of the posts, those of the organizations themselves (e.g. the amount of 1% of PIT revenue), or the parameters of the organization's Facebook profile. In the latter case, the number of followers was decidedly most important. On average, a subject's page was followed by 6,719 followers. Half of the pages had fewer than 760 followers. The disparity between the average and the median is therefore sizeable. The large dispersal of values is also reflected in a very high standard deviation (SD = 55269.54). Two of the studied profiles had more than 1 million followers, one among the large subjects, and one among the middling ones. A fourth of organizations had pages followed by fewer than 321 users, and another 25% —by more than 1,924. The size of the organization significantly correlated with its number of followers on Facebook. Both the average number of followers and the median grew from group-to-group by organizational size. For instance, in the group of largest subjects, the average number of followers was over 43 thousand, while among the smallest subjects the number was 854 (an over 50-fold difference).

User reaction (dependent variable) was measured as the number of likes, shares, and comments of each post published by the organization during the selected 14 days. A total of 4,621 posts were published, each of which was liked on average 46.52 times. The average number of shares and comments was pronouncedly lower and reached respectively 18.89 and 13.71. Total user engagement, measured as the sum of the three reactions, reached 79.13 per post on average. All three reactions displayed high values of standard deviation, which equaled 516.89 for total user engagement. In other words, the levels of engagement with the posts were highly varied, from those which prompted no reactions, to those which elicited a high number of all three types of reactions. The former of these were much more common, as evidenced by values of skewness (right-skewness means that a great majority of posts had fewer reactions than the arithmetic mean).

First, the factors influencing the probability that a particular post will be liked were analyzed (Model 1). The model involved seven predictors, of which the number of followers of a given profile proved most significant, explaining 31.6% of the variance (p<0.001). The B coefficient took positive values in this case, so as the number of followers increased, the increase in the "liking" level was also noticeable. The remaining six predictors jointly explained only 8.4% of the variance. Those predictors were: the number of links, basic post function–community building, attached image/video, amount of 1% of PIT revenue, basic post function–call to action, and day of publication–with the latter being the weakest predictor of dependent variable value. Ultimately, the constructed model was able to predict that a higher number of reactions is tied to a larger number of followers of the organization's Facebook page, as well as a smaller number of links in the post (negative B coefficient value), the basic function of the post being community building or call to action, the post including an image or video, the amount of 1% of PIT contributions, and it being published on the weekend. The summary of the analysis is presented in Table 2.

Regression analysis performed for posts by the smallest subjects (group IV) identified four predictors explanatory of the dependent variable. These were, in order: the number of followers of the organization's Facebook profile (the higher, the more likes of a post), the number of links in the post (the higher, the fewer likes), the post's basic function–community building and including an image or video in the post. These predictors jointly explained 38.3% of the variance in numbers of likes per post, of which the profile's number of followers had the

**Table 2. A multiple regression analysis for factors affecting the average number of likes, shares, comments and total engagement.**

| Elements | Model 1 | Model 2 | Model 3 | Model 4 |
|---|---|---|---|---|
| | **B** | **B** | **B** | **B** |
| | **(SE)** | **(SE)** | **(SE)** | **(SE)** |
| | **($R^2$)** | **($R^2$)** | **($R^2$)** | **($R^2$)** |
| (Constant) | -0.72*** | -1.11*** | -0.95*** | -0.72*** |
| | (0.05) | (0.05) | (0.04) | (0.04) |
| Number of followers | 0.47*** | 0.41*** | 0.34*** | 0.51 |
| | (0.01) | (0.01) | (0.01) | (0.01) |
| | (0.316) | (0.271) | (0.193) | (0.337) |
| Number of links in post | -0.64*** | -0.12** | -0.30*** | -0.65*** |
| | (0.04) | (0.04) | (0.05) | (0.04) |
| | (0.367) | (0.295) | (0.204) | (0.379) |
| Post basic function- *community building* | 0.21*** | | 0.08*** | 0.18*** |
| | (0.02) | | (0.02) | (0.02) |
| | (0.391) | | (0.209) | (0.392) |
| Post including image or video | 0.15*** | 0.06* | | 0.14*** |
| | (0.03) | (0.03) | | (0.03) |
| | (0.397) | (0.295) | | (0.396) |
| Revenues from 1% of personal income tax | 0.02** | 0.04*** | 0.03*** | 0.03*** |
| | (0.01) | (0.01) | (0.01) | (0.01) |
| | (0.398) | (0.291) | (0.212) | (0.399) |
| Post basic function- *call to action* | 0.05** | 0.14*** | | 0.08*** |
| | (0.02) | (0.02) | | (0.02) |
| | (0.399) | (0.284) | | |
| Day of the week of publication | 0.03* | -0.05** | | |
| | (0.02) | (0.02) | | |
| | (0.400) | (0.294) | | |
| Number of mentions | | -0.15*** | | |
| | | (0.04) | | |
| | | (0.293) | | |
| Number of hashtags | | | -0.11*** | |
| | | | (0.04) | |
| | | | (0.213) | |
| $R^2$ | 0.400 | 0.295 | 0.213 | 0.402 |
| *F* of change | 440.60*** | 277.74*** | 251.32*** | 517.98*** |

*Model 1*- The average number of likes; *Model 2*—The average number of shares; *Model 3*—The average number of comments; *Model 4*—The average level of total engagement

*—$p < 0.05$

**—$p < 0.01$

***—$p < 0.001$

Source: Own data

highest predictive power (18.6%). Among the subjects with revenues from PLN 100 thousand to 1 million, regression analysis yielded seven predictors of the dependent variable: the number of the profile's followers (the higher, the more likes of a post), the number of links (the higher, the fewer likes), the post's basic function–community building, including an image or video in the post, the absence of hashtags, a higher number of mentions, and the post's basic function–

call to action. Jointly, these predictors explained 27.3% of the variance in the number of likes on the post, of which the profile's number of followers had the highest predictive power (19.0%). Among the subjects with revenues from PLN 1 million to 10 million, five predictors of the dependent variable emerged. These were, in order: the number of followers of the organization's Facebook profile (the higher, the more likes of a post), the number of links (the lower, the more likes), the post's basic function–informing (negatively correlated), the post's basic function–community building, and the revenue of the organization from 1% of PIT (the higher, the more likes). These predictors jointly explained 32.3% of the variance in numbers of likes per post, of which the profile's number of followers also had the highest predictive power (23.7%). Among the subjects with the highest revenues (above PLN 10 million), seven predictors of the dependent variable emerged. These were, in order: the number of followers of the organization's Facebook profile (the higher, the more likes of a post), the number of links (the lower, the more likes), the presence of an image or video in the post, the post's basic function–informing (negatively correlated), the revenue of the organization from 1% of PIT (the higher, the more likes), the post's basic function–community building, and the number of mentions. These predictors jointly explained 57.0% of the variance in numbers of likes per post, of which the profile's number of followers again had the highest predictive power (48.1%).

Second, the factors influencing the probability that a particular post will be shared were analyzed (Model 2). In this case as well, seven predictors for the model were postulated, of which the number of followers of a profile again played the main part, explaining 27.1% of the variance (the value of B coefficient was positive). The influence of the other six predictors on the dependent variable was very small. Ultimately, the constructed model is able to predict that a higher number of reactions is tied to a larger number of followers of the organization's Facebook page, the basic function of the post being a call to action, the amount of 1% of PIT contributions, a smaller number of mentions, the post being published on a weekday, a smaller number of links and an attachment of image or video (Table 2).

Among the smallest subjects (group IV), regression analysis identified seven predictors explanatory of the dependent variable, i.e. the number of followers of the organization's Facebook profile (the higher, the more shares of a post), the number of links (the lower, the more shares), the post's basic function–call to action, the organization's revenue from 1% of PIT (the higher, the more shares), the attachment of image or video in the post, the basic function of the post–community building, and day of publication (more shares of weekday posts). These predictors jointly explained 26.7% of the variance in numbers of shares per post, of which the profile's number of followers had the highest predictive power (18.2%). Among the subjects with revenues from PLN 100 thousand to 1 million, four predictors of the dependent variable emerged, in order: the number of the profile's followers (the higher, the more shares of a post), revenue from 1% of PIT (the higher, the more shares), the post's basic function–call to action, and day of publication (more shares of weekday posts). These predictors jointly explained 25.2% of the variance in numbers of shares per post, of which the profile's number of followers had the highest predictive power (22.5%). Among the subjects with revenues from PLN 1 million to 10 million, five predictors of the dependent variable emerged. These were, in order of power: the number of the profile's followers (the higher, the more likes of a post), the post's basic function–call to action, revenue from 1% of PIT (the higher, the more shares), the number of mentions (the fewer, the more shares), and the number of hashtags (the higher, the more shares). These predictors jointly explained 21.7% of the variance in numbers of shares per post, of which the profile's number of followers had the highest predictive power (19.4%). Among the largest subjects (with revenues above PLN 10 million), four predictors of the dependent variable emerged. These were, in order: the number of the profile's followers (the higher, the more likes of a post), the number of mentions (the fewer, the more shares), the

post's basic function–informing (negative correlation), and revenue from 1% of PIT (the higher, the more shares). These predictors jointly explained 42.6% of the variance in numbers of shares per post, of which the profile's number of followers had the highest predictive power (39.2%).

Third, the factors influencing the probability that a particular post will be commented upon were analyzed (Model 3), entering five predictors explaining the dependent variable. These were, in order: the number of the profile's followers, the number of links in the post, the post's basic function–community building, revenue from 1% of PIT, and the number of hashtags in the post. These predictors jointly explained 21.3% of the variance. As illustrated in Table 2, ultimately the model showed the number of followers of the profile to have the highest predictive power (19.3%). In turn, the number of hashtags was the weakest predictor of the dependent variable. The constructed model predicts that the number of comments on a post is influenced by a higher number of followers of the profile, a smaller number of links in the post, the post's basic function–community building, higher revenue from 1% of PIT, and a smaller number of hashtags in the post.

Regression analysis by size of organization yielded similar results as in the case of likes and shares. In each of the four analyzed groups, the number of the profile's followers explained the highest percentage of the variance in the number of comments. The influence of other predictors, i.e. the post's basic function–community building, revenue from 1% of PIT (the higher, the more comments), the post's basic function–call to action, inclusion of mentions and hashtags (the fewer, the more comments), was markedly lower.

Finally, a multiple linear regression was performed to identify the factors influencing the levels of total user engagement with contents posted on Facebook (Model 4). The model included six predictors explaining total engagement level. These were, in order: the number of the profile's followers, the number of links in the post, the post's basic function–community building, inclusion of image/video, the organization's revenue from 1% of PIT and the post's basic function–call to action. Jointly, these variables explained 40.2% of the variance in the post's total engagement. As illustrated in Table 2, the final model again pointed to the number of followers as explanatory of the highest percentage of the variance (33.7%). In turn, the inclusion of image/video, the amount of revenue from 1% of PIT, and the basic function of the post–call to action were the weakest predictors of the dependent variable. Ultimately, the constructed model predicts that higher levels of Facebook user engagement with posted contents are influenced by a higher number of the profile's followers, a smaller number of links in the post, the post's basic function–community building or call to action, inclusion of image/video and higher revenue of the organization from 1% of PIT.

Multiple regression analysis yielded similar results in each of the four total annual revenue groups. In each of them, the number of the profile's followers explained the highest percentage of the variance; in the case of the smallest subjects– 21.9% (jointly, all predictors explained 38.8% of the variance), for group II subjects– 24% (jointly, all predictors explained 29.7% of the variance), for group III subjects– 25.8% (jointly, all predictors explained 32.3% of the variance), and for the largest subjects–as much as 48.3% (jointly, all predictors explained 55.9% of the variance). The influence of other predictors, i.e. the post's basic function of community building or call to action, revenue from 1% of PIT (the higher, the more total engagement), the number of links (the lower, the more total engagement), inclusion of image or video, or the number of mentions or hashtags (the lower, the more total engagement), was marginal.

The hypotheses number 3, 4, and 5 were confirmed. Firstly, the basic function of posts significantly influences the public engagement level, especially the one related to community building. Secondly, the number of followers has a statistically significant impact on public engagement. As the total number of Facebook followers increases, public engagement for

individual posts also increased. Thirdly, the number of followers of an organization's Facebook profile has more impact on public engagement with posted content than the organization's size measured as total annual revenue, its revenue from 1% of PIT, or its employment level.

On the other hand, hypothesis H2, assuming that a more complex post structure correlates with greater social media public engagement, was rejected.

## Discussion

The study results presented in the previous section carry important consequences both theoretically and practically, since they show not only whether non-governmental organizations use social media in their operations, but how they use them. At first glance, the research results allowed framing similar conclusions as in the case of the previously conducted research on Twitter and its role in building relationships with stakeholders of Polish public benefit organizations [17]. There are, however, some substantial differences. First, only in one case was Twitter employed as the major social media site, rejecting all the rest. In the remaining cases, the organization, apart from Twitter, always used Facebook. If we additionally take into account the fact that the content on Twitter was characterized by a significantly lower reaction from the public (the number of likes, shares and comments was several or over a dozen times smaller than in the case of Facebook), its role in the case of Polish public benefit organizations is at best auxiliary. It even seems that due to numerous organizational limitations related to the shortage of qualified staff, lack of time and budget shortages, especially in the case of small organizations, it is impossible to effectively run more than one social media site. The only advantage of the organizational use of Twitter is the structure of posts in terms of their basic function. The content on Twitter was mainly informative inherently inducing a minimal response from the public, because it does not serve this purpose. In the case of Facebook, the structure of analyzed posts in terms of the basic function was clearly different. The distribution of information, community building and call to action posts was very even. In this case, running two types of social media sites makes sense, however, due to a number of the above-mentioned barriers (human, financial and time), it is limited primarily to the largest entities. The research result prove that "free" technologies such as social media cannot be treated as an equalizer for smaller organizations in reaching an audience. Despite their seeming low entry barriers, effective social media use has more in common with other information technology investments. It means that the bigger the entity, the better. Thus, studies that show that smaller organizations tend to be more reactive and agile in adopting new information technologies cannot be confirmed [43]. Moreover, when we take into consideration only Facebook research, this thesis cannot be confirmed. On the one hand, some subjects were very active, on the other hand, the Facebook pages of a large portion of organizations were completely inactive throughout the study, or exhibited minimal activity. The largest organizations by total annual revenue were characterized by markedly higher posting frequency. Large organizations, therefore, seem more aware of the consequences of maintaining an inactive profile. Such profiles primarily involve damage to their brand image, since if a user visits such a profile, their relationship with the organization will most probably be marred by distrust and the conviction that this subject is not capable of pursuing its stated goals effectively. An organization can have problems with changing such a negative image. A lack of posting activity on Facebook also gives an insight into the way of thinking of the persons entrusted with the page's day-to-day operation. In the smallest organizations, such a person is probably the president of the management board, who has a number of other duties to perform. In such organizations, the financial barrier precludes hiring qualified personnel to manage this communication channel. The decision by the president of the board to create an organization's page on Facebook

was therefore probably the result of his or her fleeting interest in social media. The costless operation also probably encouraged him or her to use social media in the organization's activities. However, free operation is only an illusion, since it requires much time from the person managing these services, which cannot be devoted to other tasks. Preparing contents for publication on Facebook is no simple task, since a text of low utility will not prompt public reaction which is desirable for the organization. The person managing social media can make use of content published by other entities and simply share them by way of their page, but that too requires time and involvement. Such a situation can lead to fatigue, exhaustion with managing the profile, and its eventual abandonment to inactivity. Should this occur, it is definitely better to shut down such a profile than to leave it completely inactive, since image losses can prove difficult to repair.

The results of this study indicate that most Facebook users who come into contact with the pages of Polish PBOs are passive users. Posted content is characterized by low numbers of likes, almost 2.5 times fewer shares, and almost 3 times fewer comments. Total public engagement, measured as the sum of these three reactions, was therefore also low. The significantly higher number of likes also indicates that the behavior of the public towards the contents posted by an organization may fall into the category of "slacktivism". This term refers to political or social activism which brings no appreciable practical effects, but only bolsters the self-satisfaction of the user. Actions which are considered "slacktivism" are usually very simple and require no significant engagement from the participants. For example, by clicking the "like" button on an organization's Facebook page, the user expresses approval for a particular initiative, but their engagement is limited to that action alone, bringing no real value to the organization. The only benefit is to the user, who receives self-satisfaction in the most facile way possible. "Slacktivism" thus assumes that people who seemingly support the operations of an organization by performing very simple actions are not truly engaged or capable of sacrifice for the sake of desirable effects.

The numbers of likes, shares, and comments about content posted on Facebook were characterized by high standard deviations. Reactions to posts were therefore highly varied, with those that evoked no reactions with users, as well as those which met with exceptionally large feedback. Those "superposts" are difficult to explain, although the most likely cause of their popularity seems to be their subject matter, which usually elicits extreme emotions (e.g. a description of animal suffering). The second important condition for prompting such a reaction is a sufficiently high number of followers of the organization's Facebook page.

The results of this study allowed the formulation of a hierarchy of importance among the factors influencing public engagement with posted content. Of these, the number of a profile's followers was decidedly the first and foremost. Further factors, in order, were the post's basic function, as well as the organization's revenue from 1% of PIT, which had a markedly lesser impact on public engagement. An even less significant factor was the time of publication. The use of available communication tools rendered atypical results. Firstly, their utilization had an extremely small influence on users. Secondly, their deployment in many cases diminished user reaction levels–such a situation was observed in the case of external page links and, to a lesser extent, in the case of hashtags and mentions.

The causal relationship between the number of a profile's followers and the level of public engagement with posted contents was easy to discern. As the number of followers increased, so did the number of likes, shares, and comments. The maximization of a follower base is therefore a key challenge which stands before the management of Polish public benefit organizations. That is because the effectiveness of the use of the social media site Facebook in managing relationships depends primarily on that very factor. The larger the follower base, the larger the likelihood of attracting so-called Social Media Influencers (SMIs), i.e. individuals or

organizations which can significantly impact the perception of the organization by its stakeholders. They represent the third, most desirable element in Dhanesh's engagement model [37]. Thanks to their activities, an organization becomes more visible in the network, and can more easily reach an audience with a message. Tools which an organization may use in attracting the greatest possible number of users interested in following its posting activity can be divided into two groups. First, it can simply publish more and take advantage of the site's available communication tools. Second, it can attempt to grow its pool of followers by using traditional public relations tools, other online communication channels and static elements of its own Facebook profile. The results of the present study indicate that the order in which those instruments are deployed is of key importance. If an organization sets out to gain more followers by simply posting more and employing various communication tools in its posted content (e.g. mentions or hashtags), such attempts will be ineffective, since the likelihood of a post making a breakthrough online while the profile is followed by few users is minimal. Whatever its stylistic sophistication or wealth of communication tools, it will not have a great impact. It is much more worthwhile to first use the tools from the second group. These traditional means can include advertising leaflets delivered by mail, which, apart from information about conducted programs, may also contain details about the social media presence. A similar function may be performed by billboards, which may include the characteristic Facebook icon and the organization's page address below. An organization may also use other online channels to grow its follower base, for example, its own website, which can easily offer to redirect visitors to the Facebook page. Other social media can be used in a similar way. A useful way to attract a greater number of followers may also be the principle of reciprocity. If an organization's Facebook profile starts following the profiles of other individuals and organizations, there is a large probability that this will be reciprocated.

## Conclusions

The study results indicate that, in spite of social media's unquestionable advantages, public benefit organizations take only a small advantage of their potential. The role of Facebook in increasing an organization's revenues from 1% of personal income tax deductions seems secondary at best (a unique privilege for public benefit organizations). Such results may be surprising, especially considering the fact that the stakeholders of such organizations are significantly geographically dispersed since they are primarily individuals. Hence, reaching these persons with the organization's message is very difficult. Each additional channel of communication should be of great interest to PBOs. Neglect in this area means that holding a PBO status does not matter much to these organizations, and they might as well be ordinary registered associations or foundations.

The study encountered some limitations. The first was the fact that it did not assess the emotional charge of each published post. The study only considered the average number of comments. It also did not diagnose whether the comments were positive, negative or neutral (this aspect might help elucidate the nature of relationship-building). Further research should also focus on the composition of follower groups since proper segmentation allows content to be better suited to the requirements of the user. Finally, qualitative studies would be a fine addition to the quantitative ones. They would facilitate research on the role of Facebook in relationship-building and should be directed at social media page administrators.

## Author Contributions

**Conceptualization:** Marian Olinski, Piotr Szamrowski.

**Data curation:** Marian Olinski, Piotr Szamrowski.

**Formal analysis:** Marian Olinski, Piotr Szamrowski.

**Investigation:** Marian Olinski, Piotr Szamrowski.

**Methodology:** Marian Olinski, Piotr Szamrowski.

**Project administration:** Marian Olinski, Piotr Szamrowski.

**Resources:** Marian Olinski, Piotr Szamrowski.

**Software:** Marian Olinski, Piotr Szamrowski.

**Supervision:** Marian Olinski, Piotr Szamrowski.

**Validation:** Marian Olinski, Piotr Szamrowski.

**Visualization:** Marian Olinski, Piotr Szamrowski.

**Writing – original draft:** Marian Olinski, Piotr Szamrowski.

**Writing – review & editing:** Marian Olinski, Piotr Szamrowski.

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
