## [Decision Letter · Decision Letter 0]

29 Mar 2021

PONE-D-21-03999

Facebook as an engagement tool. How public benefit organizations are building relationships with their public?

PLOS ONE

Dear Dr. Olinski,

Thank you for submitting your manuscript to PLOS ONE. After careful consideration, we feel that it has merit but does not fully meet PLOS ONE’s publication criteria as it currently stands. Therefore, we invite you to submit a revised version of the manuscript that addresses the points raised during the review process.

The paper requires a MAJOR REVISION in order to be considered for a publication. The paper should be revised in order to follow the suggestions provided by reviewers. In particular, authors should highlight the novelty of this paper considering the previous publication, and revise the text because is too similar to the previous one. Furthermore, the methodology is basic. I suggest to the author to add other methodologies, as required by reviewers.

We look forward to receiving your revised manuscript.

Kind regards,

Barbara Guidi

Academic Editor

PLOS ONE

Journal Requirements:

2. During your revisions, please confirm whether the wording in the title is correct and update it in the manuscript file and online submission information if needed. Specifically, whether "Facebook as an engagement tool. How public benefit organizations are building relationships with their public?" should be changed to "Facebook as an engagement tool: how are public benefit organizations building relationships with their public?

"The funders had no role in study design, data collection and analysis, decision to

publish, or preparation of the manuscript."

Reviewers' comments:

Reviewer's Responses to Questions

**Comments to the Author**

1. Is the manuscript technically sound, and do the data support the conclusions?

Reviewer #1: Partly

Reviewer #2: Yes

Reviewer #3: Partly

2. Has the statistical analysis been performed appropriately and rigorously? 

Reviewer #1: No

Reviewer #2: Yes

Reviewer #3: Yes

3. Have the authors made all data underlying the findings in their manuscript fully available?

Reviewer #1: No

Reviewer #2: No

Reviewer #3: No

4. Is the manuscript presented in an intelligible fashion and written in standard English?

Reviewer #1: No

Reviewer #2: Yes

Reviewer #3: Yes

5. Review Comments to the Author

Reviewer #1: Thanks for submitting the manuscript. However, there are several major flaws in this paper which I do not think it is suitable for publication. You may refer to my comments as follow:

1. The research paper is lack of scientific rigor. The literature review section has been highlighted many theoretical perspectives (p. 3-9). The authors may need to be more specific and mainly focus on the theory that is relevant to this study. Also, none of the hypothesis (p. 10-13) are related to those perspectives. There is lack of linkage between the discussion section and the theory highlighted in the literature review section. These sections may require significant amount of work to rewrite it.

2. Another major issue is related to the relevancy of the public benefit organization (PBO) with the rest of the world. The specific government policy in Poland which enable the PBO to receive financial support via 1% of personal income tax (p. 12) and some countries did not allow PBO to use social media platform (p. 3). I am afraid that the findings of this paper maybe more suitable for those region/country specific journal.

3. The authors mentioned they have got a recent publication ‘Twitter as an engagement tool: How public benefit organizations are building relationships with their audience’ in Journal of East European Management Studies (Olinski & Szamrowski, 2020) on p. 12 (line 286). The current submission ‘Facebook as an engagement tool. How public benefit organizations are building relationships with their public?’ is just reporting very similar findings. What is the novelty and contribution of this study? There is also a potential danger of text recycling or salami slicing.

4. The data collection method is also not scientifically sound. With the help of AI and data mining technique, the authors should collect the data from all the PBOs rather than just focused on ‘876 subjects’ (p. 15, line 349). The operationalization of major variables is not clear.

5. The procedure for analysis is unclear, which specific mathematical algorithm was employed (p. 16, line 382)? Any sources or justifications?

6. The entire manuscript with many irrelevant contents, especially in the literature review and methods, and results sections. The authors should present the ideas and findings concisely. There are also significant number of typos and grammatical mistakes. They authors are highly recommended to proof read the manuscript, preferably by English native speaker, prior to the submission.

Reviewer #2: This is an interesting study and I would recommend publication after minor revisions (as outlined below), are made.

1. The used methods are not clear in the abstract. Please add a couple of lines about the methods there.

2. The analyses are appropriate to test the stated hypotheses, and the results are interpreted correctly. However, more clarity would improve the results section. Please see a list of points to be improved below.

a) H1 should be shorter. I suggest only keeping the second part. As it is stated now it sounds like two hypotheses. In addition, you haven't specified what constitute "low" publishing activity. Without a point of comparison, it does not make sense to talk about "high" or "low" activity.

b) Overall, results are presented in order, that is, each hypothesis in tested one after the other, which is good. However, the flow and clarity would improve if the authors could add whether each hypothesis was supported or not at the end of each section, instead of having this all at the end of the results.

c) Some results could be removed, as they do not test any hypotheses. The results section is already long winded, so shortening it would improve clarity. For example, the results reported between the testing of H1 and H2, at lines 498 - 527, could be removed.

d) Table 2 reports significance levels, which is good. However, it would be good to report those in text too. When reporting the testing of H2, the authors stated, at line 542, that "Organization size had a significant influence on the frequency of use of contents". However, the reader need to scroll all the way down to Table 2 to find evidence of this significance. Same goes for "most significant" at line 590.

e) H3 is not stated very clearly. The authors need to specify which function is expected to create more/which type of public engagement.

f) H4 needs to be split into two hypotheses.

3) Some additional proofreading should be done. E.g. the first person is sometimes used (such as at line 528).

4) In relation to data availability, the authors declare that "The data underlying the results presented in the study are available from social media platforms and public benefit organizations financial reports (everything is widely available on the Internet)". However, I assume that the authors need to submit the actual SPSS dataset they created either to a public depository or as additional material to PLOS ONE.

Reviewer #3: In this paper the authors present a study of the Facebook page owned by a set of Polish public benefit organizations (PBO). The analyses show that pages with a larger number of users tend to be more active, make the followers interact more, and receive a larger share of public founding. The paper is well written, but the analyses are rather basic.

In this paper i see two major shortcomings:

- The scenario of Polish PBO is very specific. Is there a reason to target only one Nation? And why only the Polish ones? Are the results similar to PBO of other Nations? Why is this scenario in particular extremely relevant? Overall the scenario is unnecessarily very focused. I expect to see a detailed explanation of this choice.

- The analyses performed are rather basic. Number of posts, number of reactions, basic statistics are the used tools to prove the hypotheses. Did you check that the interactions are legit and not produced by bot? Are these interactions socially relevant? Are the interactions positive or negative? What is the audience reached by the pages? I think that a more detailed study is due to confirm the hypotheses

Additionally, I point out a few other unclear points:

- Page 15 lines 348-351: why do you sample the population? with such a small number of samples you can just analyse all the data you have instead of reducing the dataset by 60%

- Page 15, lines 355-359: In the data collection section, I think that the part where you consider the randomly selected days of the week should be explained better, because I did not get what you mean there.

- A proofreading round should eliminate the few typos present in the paper.

6. PLOS authors have the option to publish the peer review history of their article (what does this mean?). If published, this will include your full peer review and any attached files.

Reviewer #1: **Yes: **Sai-fu Fung

Reviewer #2: No

Reviewer #3: No

---

## [Author Response · Author response to Decision Letter 0]

17 May 2021

Answers to Editor

1. Manuscript meets PLOS ONE’s style requirements (according to The PLOS ONE style template)

2. We have changed the title of the manuscript as suggested: Facebook as an engagement tool: how are public benefit organizations building relationships with their public? 

3. All relevant data are within the manuscript.

 We want to change the Data Availability statement. There is no repository information for our data at acceptance (it is explained in the cover letter). 

4. Financial statements

a) The research did not receive any financial or material support.

b) We state: The funders had no role in the study design, data collection and analysis, decision to publish, or preparation of the manuscript.

c) The authors did not receive any salary

d) The authors received no specific funding for this work.

Answers to reviewers

Reviewer 1

1. The structure of the theoretical part has been improved. The literature was associated with the research part of the work - especially with hypotheses.

2. Some results described in the paper have a universal character and they may apply regardless of the country. Legal solutions used in Poland force organizations with PBO status to communicate with an individual to encourage them to donate funds to the organization. Despite the lack of such solutions in other countries, the behavior of these organizations is similar, as they also have to communicate with people from their environment in order to raise funds.

3. This article is part of a larger research project that analyzed the use of social media in the activities of non-profit organizations. The conclusions from the use of Twitter and Facebook clearly differ. Only similar research procedures were used, while the conclusions were different. Twitter builds engagement with the public in a different way than Facebook, and its role in Polish organizations is much smaller (it does not have the same application as in the case of Anglo-Saxon countries).

4. The research sample was selected in a traditional manner and in accordance with the rules of statistics (stratified sampling). The methods of data collection using artificial intelligence do not provide the possibility of full analysis. The manual analysis made it possible to select the organizations that will be the case studies in subsequent publications. Manual research allowed to identify all the nuances of using Facebook in the surveyed organizations. In addition, obtaining financial data about the organization was not possible with the use of artificial intelligence.

5. Corrected.

6. The comment was taken into account in full. The literature part has been shortened. Only relevant fragments were left (ideas and findings were presented concisely). The text was checked once more by the company providing services to the University of Warmia and Mazury in Olsztyn (attached confirmation) and submitted to a native speaker (Michael Thoene) for checking.

Reviewer 2

1. Corrected (specific information on methods added).

2. According to the list of points (a-f)

2a. corrected - only the second part of the hypothesis remained,

2b. corrected - hypothesis verification is placed under each section,

2c. corrected - some results that did not test any hypotheses were removed (The fragment 528-560 was deleted). We left fragment 498-527 (the lack of communication tools in posts would make it impossible to confirm or reject hypothesis 2),

2d. corrected in accordance with the reviewer's comment,

2e. corrected,

2f. corrected (H4 was split into two hypotheses H4 and H5).

3. Improved throughout the article 

Reviewer 3

1. This article brings the following contribution to the scholarship literature. To begin, it builds on the effectiveness of computer-mediated organizational communication. This is the first organizational-level study of Facebook effectiveness with a unique type of nonprofit organization with the ability to receive 1 % of personal income tax. The Polish case may be a valuable example for the implementation of similar solutions for nonprofit organizations in other countries. The growing importance of social media in the activities of nonprofit organizations requires taking steps to create effective strategies for their use. This is especially important concerning limited resources that usually nonprofit organizations own. We consider "online engagement" as a vital intermediate objective, which should result in real world activities by the organizations’ online audience (e.g. do‐ nation, voluntary work, lobbying).Therefore, it can be concluded that despite the increasing technological advancement and the use of sophisticated information exchange techniques, the organization itself and the ideas it presents are still the most important (this can later be translated into content published in online channels). It is important, therefore, that a given idea and related contents arouse controversy, evoke strong emotions, are up to date, etc. As an example, you can give an action on Facebook "Stop lawlessness" regarding the reform of the Polish judiciary and related EU objections regarding the state of law in Poland. This problem caused much controversy in Poland and "the real world" protests were current, that's why posts associated with this event attracted a large crowd of followers, and consequently triggered a large audience reaction. Another example are the posts about extreme animal suffering, which can also trigger a similar reaction. The use of sophisticated communication tools in this case, such as a hashtag, mention or hyperlinks, was not very important and only fulfilled an auxiliary role. Thus, it can be seen that social media, despite different ways of disseminating information (compared to traditional media), are still grounded in real problems, and the number of followers (observers) does not depend on technological advancement in the way information is provided (e.g. post construction), only from the "real" problem. This principle seems to be universal, i.e. it applies regardless of the country, because human behaviour in this respect is rather similar.

2. Interactions were checked. The entire research was done by hand, so with a high degree of probability it can be said that they were not bots. We have not studied the nature of the interactions (whether they are positive or negative). Audience analysis will be the subject of separate studies.

3. The research sample was selected in a traditional way and in accordance with the rules of statistics (stratified sampling). Therefore, not all the organizations from group 1 were examined, despite their small number (64 entities out of 77).

4. Page 15 – line 355-359, corrected.

5. Eliminated typos. Forwarded to proofreading done by a native speaker.

---

## [Decision Letter · Decision Letter 1]

9 Jun 2021

PONE-D-21-03999R1

Facebook as an engagement tool: how are public benefit organizations building relationships with their public?

PLOS ONE

Dear Dr. Olinski,

Thank you for submitting your manuscript to PLOS ONE. After careful consideration, we feel that it has merit but does not fully meet PLOS ONE’s publication criteria as it currently stands. Therefore, we invite you to submit a revised version of the manuscript that addresses the points raised during the review process.

The paper needs a MAJOR REVISION. The main issue highlighted by reviewers is the similarity of this work with another. Authors should improve the paper in order to address this important issue in order to be evaluated for a publication

We look forward to receiving your revised manuscript.

Kind regards,

Barbara Guidi

Academic Editor

PLOS ONE

Reviewers' comments:

Reviewer's Responses to Questions

**Comments to the Author**

1. If the authors have adequately addressed your comments raised in a previous round of review and you feel that this manuscript is now acceptable for publication, you may indicate that here to bypass the “Comments to the Author” section, enter your conflict of interest statement in the “Confidential to Editor” section, and submit your "Accept" recommendation.

Reviewer #1: (No Response)

Reviewer #3: All comments have been addressed

Reviewer #4: (No Response)

2. Is the manuscript technically sound, and do the data support the conclusions?

Reviewer #1: Partly

Reviewer #3: (No Response)

Reviewer #4: No

3. Has the statistical analysis been performed appropriately and rigorously? 

Reviewer #1: Yes

Reviewer #3: (No Response)

Reviewer #4: No

4. Have the authors made all data underlying the findings in their manuscript fully available?

Reviewer #1: No

Reviewer #3: (No Response)

Reviewer #4: Yes

5. Is the manuscript presented in an intelligible fashion and written in standard English?

Reviewer #1: No

Reviewer #3: (No Response)

Reviewer #4: No

6. Review Comments to the Author

Reviewer #1: Thanks for submitting the revised manuscript. However, the authors still failed to address my concerns, in particular points number 1, 2, 3, 4, and 5. The revised manuscript did not made any significant changes on the highlighted issues in my previous review report.

The linkage between the theory and hypothesis are still missing. The discussion section also failed to highlight any theoretical implications.

The most serious problem is still related to the Editorial policies (point number 3). The subheadings of the Twitter paper (Olinski & Szamrowski, 2020b) are identically the same with the current submission. The hypotheses, variables used on Table 3 (p. 235), and the findings are also more or less the same. The authors still failed to address my concerns about the new insight of this manuscript. The methodology section is also with the evidences of text recycling from the authors’ other recent publications (Olinski & Szamrowski, 2020a, 2020c).

In view of this above issues, I do not think this manuscript is suitable for publication.

References

Olinski, M., & Szamrowski, P. (2020a). Stewardship Concept Utilization on the Websites of Polish Public Benefit Organizations. Romanian Journal of Communication and Public Relations, 22(2), 91-106. Retrieved from <go isi="" to="">://WOS:000563730100006

Olinski, M., & Szamrowski, P. (2020b). Twitter as an engagement tool: How Public Benefit Organizations are building relationships with their audience. Journal of East European Management Studies, 25(2), 216-245. doi:10.5771/0949-6181-2020-2-216

Olinski, M., & Szamrowski, P. (2020c). Using Websites to Cultivate Online Relationships: The Application of the Stewardship Concept in Public Benefit Organizations. Journal of Nonprofit & Public Sector Marketing, 28. doi:10.1080/10495142.2020.1798853</go>

Reviewer #3: (No Response)

Reviewer #4: In this paper the authors present a study of the Facebook page owned by a set of Polish public benefit organizations (PBO).

The analyses show that pages with a larger number of users tend to be more active, make the followers interact more, and receive a larger share of public founding.

The paper is well written, even if the analyses are really simple.

The subheadingsThe main issue of this paper is that it presents an analysis which is more or less the same of other publications that are not cited there:

- Twitter as an engagement tool: How Public Benefit

Organizations are building relationships with their

audience - Olinski & Szamrowski.

The content of Table 1 and 2 is the same of the article cited above. What is the difference? What is the improvement that this paper could give to the literature?

Authors should cite these papers and provide a clear evidence of what is the novelty of this study, and a comparison with the previous works.

Furthermore, text is also similar to other papers. Please, change all the article in order to be original.

7. PLOS authors have the option to publish the peer review history of their article (what does this mean?). If published, this will include your full peer review and any attached files.

Reviewer #1: **Yes: **Sai-fu Fung

Reviewer #3: No

Reviewer #4: No

---

## [Author Response · Author response to Decision Letter 1]

29 Jul 2021

1. We have changed the subheadings and text within them to avoid repetitions from other papers.

2. We have changed the hypothesis 2 (H2).

3. Novelty and originality of the research was underlined by adding, the following sentences to the introduction part of the article (some sentences were also moved or deleted):

Research conducted on the organizational uses of social media has primarily been concerned with commercial, for-profit entities. The scholarship literature on non-profit organizations is much sparser. This is a surprising fact, since social media can be very useful in several fields, e.g. volunteer recruitment, fundraising, media exposure and especially relationship management with the stakeholders of the organization. The literature analyses user engagement with contents published primarily on Facebook and Twitter, but the number of studies is low, since the term engagement itself is not entirely understood, which impedes the analysis (lines 58-65).

Previous research has also concentrated on large organizations, primarily based in the United States and, again, their organizational uses of Facebook and Twitter, which is especially popular in Anglo-American countries. Moreover, the researchers of these issues, when choosing a research sample, most often limited themselves to the method that can be described as "top-100-style sampling". It creates a research gap, as noted by Nah and Saxton denoting the underrepresentation of small- and medium-sized organizations in research samples. The use of random sampling is the solution to this problem, allowing more accurate inference via increased generalizability and simpler hypothesis testing. The appeal of both of these researchers only in a few cases was taken into account. Polish academic literature regarding the utilization of social media in the activities of non-profit organizations is also very modest. With a few exceptions, there has been practically no research conducted on the role of social media in building relationships with the community by non-profit organizations, especially by public benefit organizations (lines 65-78 ).

The analysis of academic literature also indicates that the vast majority of studies conducted so far have dealt with only a single social media channel. Therefore, there is a lack of research describing the coordination process of different communication channels. It seems that this issue is particularly important from a practical point of view, enhancing the stakeholder’s engagement (lines 79-83 ).

The present article aims to fill in at least part of that gap. Firstly, it contributes to the literature by examining the scope of the public benefit organizations’ use of the Facebook social network in managing relationships with the public (lines 84-86 ).

Secondly, the research described in the following article was part of a larger research project, which included an analysis of the use of Twitter in the activities of Polish public benefit organizations. The research methodology used for its analysis was then applied to Facebook research. The general conclusions formulated on this basis made it possible to identify differences in the organizational use of both communication channels and the development of practical guidelines for coordinating the utilization of different social media services in a non-profit organizations. Thirdly, random sampling was used to select the organization for the research, taking into account not only large organizations, but also medium and small organizations, meeting the appeal formulated by Nah and Saxton (lines 94-103 ).

4. Much more emphasis was placed on pointing out the relationship between literature review and hypotheses development. We added the following sentences to the literature review part of the article (some sentences were also moved or deleted). 

Resource dependency theory is a starting point in explaining the reasons for the use of social media in the activities of a non-profit organization. As indicated by Mato-Santiso, Rey-García and Sanzo-Perez “this is because of non-profit organization’s endemic lack or shortage of resources relative to the size of their target groups’ needs (potential recipients, beneficiaries or users) and to the complexity of the social problems tackled” (lines 114-119).

The research that has investigated the use of social media for PR activities of non-profit organizations has so far been focused on only a few of its aspects. First, on the scope of social media usage and on which characteristics employed by the NGO increased the use of those services. Second, static elements of an organization’s profile on the social media site were analyzed. Third, the perception of the role of social media for an NGO by public relations practitioners were examined. Fourth, the contents published on social media sites and their impact on user engagement levels were analyzed. From the perspective of this article, the fourth aspect mentioned above is the most important which is associated with building public engagement (lines 147-155 ).

Engagement is defined as a dynamic multidimensional relational concept including psychological and behavioral features of involvement, connection, interaction, and participation designed in a way to enable the achievement of goals at an individual, organization or, social levels. Engagement can be understood as a process in which engagement is created, or cocreated, through communication or as a state or outcome that encompasses cognitive, affective, and behavioral dimensions. In the online environment, the theoretical foundations for the formation of engagement can be found in the work of, for example, Dhanesh, who also understands this concept as a cognitive, emotional and behavioral state in which the organization and the entities in its environment, sharing common interests in areas relevant to themselves, enter interactions which can be placed on a scale (lines 156–165).

The literature indicates that public engagement is influenced by a number of factors that may result from both the nature of the published content and the very characteristics of a non-profit organization. The first group of factors may include, for example, the function performed by the published content Research conducted by Lovejoy and Saxton, which made it possible to create a typology enabling the identification of the basic and specific function of content published in the online environment. This particularly concerned the Twitter content of American largest non-profit organizations. Other studies indicate the importance of communication tools used in single posts or tweets. It may be assumed that the more complex the post, the more likely the recipient would react (the presence of a video file, photos, links to external sites, hashtags or mentions in published content). Moreover, the length of the post or tweet and the time of its publication, e.g. a weekday or weekend, may also translate into a greater level of public engagement. The second group of factors concerning the features of the non-profit organization itself may also influence the public engagement of the content published on social media sites. These features may include the value of the organization’s assets or annual revenues, its age, type of activity, the number of followers of its social media profile (lines 182–197 ).

Kim and Yang treat pressing a “like” button as an affect-driven behavior, commenting on content as a cognitively triggered behavior and sharing as a combination of both. For example, the recipient of the content, by clicking on the "like" button, informs that he or she has a positive attitude towards the content and identifies with it to some extent. According to Saxton and Waters, liking a single post is the weakest form of engagement, as opposed to the number of comments, which should be prioritized from an organizational point of view, reflecting a significant commitment of the content’s recipient (lines 202–208 ).

5. In the subsection "Hypothesis development" reference is made to previous own research

This assumption also results from the research conducted earlier by the authors on the use of Twitter by Polish non-profit organizations (lines 250–251 ).

The importance of this factor is also indicated by the results of the Wallace and Rutherford studies, although the research sample in this case was only 160 non-profit organizations (lines 252–254 ).

6. In the subsection "Discussion" reference is made to previous own research. We added the following text

At first glance, the research results allowed framing similar conclusions as in the case of the previously conducted research on Twitter and its role in building relationships with stakeholders of Polish public benefit organizations. There are, however, some substantial differences. First, only in one case was Twitter employed as the major social media site rejecting all the rest. In the remaining cases, the organization, apart from Twitter, always used Facebook. If we additionally take into account the fact that the content on Twitter was characterized by a significantly lower reaction from the public (the number of likes, shares and comments was several or several times smaller than in the case of Facebook), its role in the case of Polish public benefit organizations is at best auxiliary. It even seems that due to numerous organizational limitations related to the shortage of qualified staff, lack of time and budget shortages, especially in the case of small organizations, it is impossible to effectively run more than one social media site. The only advantage of the organizational use of Twitter is the structure of posts in terms of their basic function. The content on Twitter was mainly informative, inherently inducing a minimal response from the public, because it does not serve this purpose. In the case of Facebook, the structure of analyzed posts in terms of the basic function was clearly different. The distribution of information, community building and call to action posts was very even. In this case, running two types of social media sites makes sense, however, due to a number of the above-mentioned barriers (human, financial and time), it is limited primarily to the largest entities. The research result proves that “free” technologies such as social media cannot be treated as an equalizer for smaller organizations in reaching an audience. Despite their seeming low entry barriers, effective social media use has more in common with other information technology investments. It means that the bigger the entity, the better. Thus, studies that show that smaller organizations tend to be more reactive and agile in adopting new information technologies cannot be confirmed (lines 723–748 ).

---

## [Decision Letter · Decision Letter 2]

18 Aug 2021

Facebook as an engagement tool: how are public benefit organizations building relationships with their public?

PONE-D-21-03999R2

Dear Dr. Olinski,

We’re pleased to inform you that your manuscript has been judged scientifically suitable for publication and will be formally accepted for publication once it meets all outstanding technical requirements.

Kind regards,

Barbara Guidi

Academic Editor

PLOS ONE

Additional Editor Comments (optional):

Reviewers' comments:

Reviewer's Responses to Questions

**Comments to the Author**

1. If the authors have adequately addressed your comments raised in a previous round of review and you feel that this manuscript is now acceptable for publication, you may indicate that here to bypass the “Comments to the Author” section, enter your conflict of interest statement in the “Confidential to Editor” section, and submit your "Accept" recommendation.

Reviewer #1: All comments have been addressed

Reviewer #3: All comments have been addressed

Reviewer #4: All comments have been addressed

2. Is the manuscript technically sound, and do the data support the conclusions?

Reviewer #1: Yes

Reviewer #3: (No Response)

Reviewer #4: Yes

3. Has the statistical analysis been performed appropriately and rigorously? 

Reviewer #1: Yes

Reviewer #3: (No Response)

Reviewer #4: Yes

4. Have the authors made all data underlying the findings in their manuscript fully available?

Reviewer #1: Yes

Reviewer #3: (No Response)

Reviewer #4: Yes

5. Is the manuscript presented in an intelligible fashion and written in standard English?

Reviewer #1: Yes

Reviewer #3: (No Response)

Reviewer #4: Yes

6. Review Comments to the Author

Reviewer #1: Thanks for submitting the revised manuscript. I am fully satisfied with the responses and changes made by the authors. I do not have any further comments.

Reviewer #3: (No Response)

Reviewer #4: The paper has been revised as required. No further comments. All previous comments have been addressed.

7. PLOS authors have the option to publish the peer review history of their article (what does this mean?). If published, this will include your full peer review and any attached files.

Reviewer #1: **Yes: **Sai-fu Fung

Reviewer #3: No

Reviewer #4: No

---

## [Editor Report · Acceptance letter]

31 Aug 2021

PONE-D-21-03999R2 

Facebook as an engagement tool: how are public benefit organizations building relationships with their public? 

Dear Dr. Olinski:

I'm pleased to inform you that your manuscript has been deemed suitable for publication in PLOS ONE. Congratulations! Your manuscript is now with our production department. 

Kind regards, 

on behalf of

Dr. Barbara Guidi 

Academic Editor

PLOS ONE